# (*S*)-*N*-Benzyl-1-phenyl-3,4-dihydroisoqunoline-2(1*H*)-carboxamide Derivatives, Multi-Target Inhibitors of Monoamine Oxidase and Cholinesterase: Design, Synthesis, and Biological Activity

**DOI:** 10.3390/molecules28041654

**Published:** 2023-02-09

**Authors:** Qing-Hao Jin, Li-Ping Zhang, Shan-Shan Zhang, Dai-Na Zhuang, Chu-Yu Zhang, Zhou-Jun Zheng, Li-Ping Guan

**Affiliations:** 1College of Nursing, Zhejiang Pharmaceutical University, Ningbo 315153, China; 2Food and Pharmacy College, Zhejiang Ocean University, Zhoushan 316022, China

**Keywords:** 3,4-dihydroisoquinoline, carboxamide derivatives, monoamine oxidase, cholinesterase, cytotoxicity, molecular docking

## Abstract

A series of (*S*)-1-phenyl-3,4-dihydroisoquinoline-2(1*H*)-carboxamide derivatives was synthesized and evaluated for inhibitory activity against monoamine oxidase (MAO)-A and-B, acetylcholine esterase (AChE), and butyrylcholine esterase (BChE). Four compounds (**2i**, **2p**, **2t**, and **2v**) showed good inhibitory activity against both MAO-A and MAO-B, and two compounds (**2d** and **2j**) showed selective inhibitory activity against MAO-A, with IC_50_ values of 1.38 and 2.48 µM, respectively. None of the compounds showed inhibitory activity against AChE; however, 12 compounds showed inhibitory activity against BChE. None of the active compounds showed cytotoxicity against L929cells. Molecular docking revealed several important interactions between the active analogs and amino acid residues of the protein receptors. This research paves the way for further study aimed at designing MAO and ChE inhibitors for the treatment of depression and neurodegenerative disorders.

## 1. Introduction

Monoamine oxidase (MAO) and cholinesterase (ChE) play important roles in the regulation of nerve conduction by catalyzing the degradation of neurotransmitters. Along with abnormal protein aggregation, low neurotransmitter levels are associated withneuron death [1,2,3]. Alzheimer’s disease (AD) is a common neurodegenerative diseasecharacterized by low levels of the neurotransmitters acetylcholine (ACh) and butyrylcholine (BCh), both of which are involved in cognition. Although multiple hypotheseshave been proposed, the exact mechanism of AD development remains elusive; however, the cholinergic hypothesis offers the best explanation to date [4,5]. Clinical treatment of AD involves enhancing cholinergic function by prolonging the usability of ACh that is released into the synaptic space of neurons; this can be achieved byusing ChE inhibitors, which inhibit the enzyme responsible for the degradation of ACh. The inhibition of acetylcholinesterase (AChE) and butyrylcholinesterase (BChE) is considered to be beneficial in the treatment of AD [6,7,8,9].

Late-life depression (defined as depression in individuals aged sixty years and older) and neurodegenerative disorders, such as AD, are often related and may have a common underlying cause. Depression is a prodromal symptom and risk factor for AD, and may affect up to 50% of symptomatic AD patients [10,11,12]. MAO is responsible for degrading monoamine neurotransmitters through oxidative deamination. Low levels of monoamine neurotransmitters have been reported to be involved in various pathological processes, including depression and neurodegeneration [13,14]. Elevated MAO levels have been reported as a biomarker in AD patients. It has been hypothesized that elevated MAO levels induce excessive production of hydroxyl radicals in the brain, triggering a biochemical cascade that is associated with beta-amyloid plaque deposition [15,16,17,18]. Furthermore, diverse pharmacological studies have shown that MAO inhibitors have neuroprotective activity associated with the relief of oxidative stress through the regulation of mitochondrial dysfunction [19].

In recent years, the “one molecule, one target” model of drug development has beenshown to be inadequate for multi-factorial diseases, especially AD and depression. In light of this, emerging multi-target directed ligands (MTDLs) are promising candidates for the treatment of multi-factorial diseases [20,21]. Ramsay et al. [22] showed that AChE, BChE, MAO-A, and MAO-B are suitable targets for MTDLs. Therefore, ChE and MAO inhibitors have the appropriate properties for the development of MTDLs as neuroprotective and symptomatic drugs and for modulating amyloidogenic pathways.

Isoquinolines and isoquinoline alkaloids are types of plant nitrogen metabolites that have important pharmacological effects, including inhibitory effects against AChE, BChE, MAO-A, MAO-B, and beta-amyloid aggregation [23,24,25]. Wan Othman et al. [26] have reported that reticuline and nornantenine, which were isolated from the bark of *Cryptocaryagriffithiana* Wight, were moderate inhibitors of BChE, with IC_50_ values of 65.04 and 94.45 µM, respectively. Baek et al. [27] have reported that chelerythrine potently and selectively inhibited MAO-A, with IC_50_ values of 0.55 µM for MAO-A and > 20.0 µM for MAO-B. Avicine, which was isolated from a root extract of *Zanthoxylumrigidum*spp. by Gonzalez et al. [28], displayed good inhibitory activity against ChE enzymes and MAO-A, with IC_50_ values of 0.15 µM electric eel acetylcholinesterase (*Ee*AChE), 0.88 µM horse serum butyrylcholinesterase (*Eq*BChE), and 0.41 µM (MAO-A). These results suggested that avicine may be a promising MTDL candidate for the development of new therapeutic agents against AD (Figure 1). In addition, there are clinically approved MAO and ChE inhibitors with carboxamide groups, including moclobemide, toloxatone, and rivastigmine (Figure 1).

Recently, our research group has reported the preparation of a series of 3,4-dihy-droisquinoline-2(1H)-carboxamide analogs as candidate antidepressant agents. The compound (S)-*N*-benzyl-1-phenyl-3,4-dihydroisoquinoline-2(1*H*)-carboxamide (BPIQC) displayed the highest antidepressant effect in the forced swimming test (duration of immobility: 11.0 ± 7.1 s, *p*< 0.001) [29]. Because of our continued interest in the development of dihydroisoquinoline-2(1*H*)-carboxamide analogs for the treatment of AD and depression, the aim of the present study was to explore the synthesis of a series of (*S*)-1-phenyl-3,4-dihydroisoquinoline-2(1*H*)-carboxamide analogs with different *N*-benzyl substituents using BPIQC as the lead compound. We discuss the potential of the target compounds as inhibitors of AChE, BChE, MAO-A, and MAO-B. In addition, we report a structure–activity relationship (SAR) study of the derivatives, in which the compounds were assessed for cytotoxicity using the 3-(4,5-dimethylthiazol-2-yl)-2,5-diphenyltetrazolium bromide (MTT) assay andacridine orange (AO) fluorescent staining. Molecular docking was used to elucidate the binding modes of the inhibitors.

## 2. Results and Discussion

### 2.1. Design of Analogs

The designed molecules consist of three basic components: (i) an isoquinoline ring, which is a major contributor to π–π stacking; (ii) an acyclic carboxamide moiety possessing a nitrogen atom and oxygen atom thatcan act as forminghydrogen bond donor/acceptor sites for building important interactions with the amino acid residues at the active sites of the enzymes; (iii) a basic foundation of carboxamide-fragment-linked benzyl: recently, our research groupfound that compound BPIQCdisplayed the highest antidepressant effect with carboxamide-fragment-linked benzyl [29], in which designing idea is to link the benzyl of donepezil and carboxyl amide of moclobemide together (Figure 2). Therefore, the concept of hybridization not only provides a new and diversified lead structure, but also markedly maintains the pharmacokinetic parameters.Therefore, all kinds of dynamic SAR analyses can be performed to describe the impact of substituent or functional group changes on biological potential.

### 2.2. Synthetic Chemistry

A family of (*S*)-1-phenyl-3,4-dihydroisoquinoline-2(1*H*)-carboxamides containing *N*-benzyl substituents was synthesized using a simple route, as shown in Figure 1. Substitution reactions of commercially availablebenzylamine with different substituentsproduced the substituted isocyanates **1a**–**1ac**, which under wentnucleophilic substitution reactions using triphosgene in 70–85% yield. Subsequently, the substituted isocyanates were condensed with (*S*)-1-phenyl-1,2,3,4-tetrahydroisoquinoline under triethylamine catalysis to afford the desired target compounds **2a**–**2ac**. Structural characterization and purity determination of the compounds were performed using FT-IR, ^1^H-NMR, ^13^C-NMR, ESI-HRMS, and HPLC. (see Appendix A)

### 2.3. Spectroscopic Characterization

The peaks of benzene ring protons, relying on their chemical environment, appeared at approximately 6.82–7.56 ppm. The C-NH signal emerged comparatively upfield as a triplet (*J* = 9.0 Hz or *J* = 6.0 Hz) in the compounds. The triplet spin multiplicity of the –NH peak is ascribed to the coupling of this proton possessing –CH_2_ protons in its immediate vicinity. The upfield chemical shift may be considered to be the alkyl carbon atom succeeding –NH that shields the proton [8]. The methine group linked to the benzene ringproton (Ph-CH) displayedan independent single peak downfield at 6.3–6.4 ppm. The ^13^C-NMR spectra further proved the new structures bythe presence of a C=O carbon peak at approximately157.31–162.61 ppm andthe presence of signals for the 1,2,3,4-tetr-ahydroisoquinoline ring carbon atoms (for compound **2v**, these appeared at 28.36 (CH_2_), 40.16 (NCH_2_), 40.76 (NCH_2_CH_2_), and 57.88 (Ph-CH) ppm (Figure 3).

### 2.4. In Vitro Biological Evaluation

#### 2.4.1. Inhibitory Activity of Compounds **2a**–**2ac** on MAO

The newly prepared (*S*)-1-phenyl-3,4-dihydroisoquinoline carboxamide derivatives **2a**–**2ac** were tested for inhibitory activity against MAO-A and MAO-B enzymes. First, a preliminary concentration screen of the 29 derivatives **2a**–**2ac** was conducted (Table 1). Nineteen compounds exhibited inhibitory activity against MAO, with inhibition rates ranging from 29.2 to 71.8% at 100 µM. Among them, six compounds (**2d**, **2i**, **2j**, **2p**, **2t**, and **2v**) showed inhibition rates of more than 50% (52.0–71.8%), and compound **2d,** with a *para*-F substituent, displayed the highest inhibitory activity at 71.8%. Next, six compounds (**2d, 2i, 2j, 2p, 2t**, and **2v**) were chosen for the determination of IC_50_ values. Derivatives **2i**, **2p**, **2t**, and **2v**, incorporating *meta*-Br, *ortho*-CH_3_, *meta*-OCH_3_, and 2,4-(OCH_3_)_2_ groups on the benzyl ring, respectively, effectively inhibited both MAO-A and MAO-B; however, their inhibitory activities were lower than that of the drug Rasagiline. In contrast, compounds **2d** and **2j**, with *para*-F and *para*-Br groups on the benzyl ring, respectively, selectively inhibited MAO-A, with IC_50_ values of 1.38 and 2.48 µM, respectively. Selective MAO-A inhibitors are suitable for the treatment of anxiety and depression [30], whereas MAO-B inhibitors are suitable for the treatment of Parkinson’s disease (PD) and AD [31]. Many selective inhibitors for MAO-A and MAO-B have been reported, and some are now used to treat neurological disorders in the clinic [32,33].

#### 2.4.2. In Vitro Cholinesterase Inhibition of Compounds **2a**–**2ac**

Carboxamide compounds **2a**–**2ac** were screened for their ability to inhibit AChE and BChE in vitro using Ellman’s method (Table 2) [34]. Donepezil and tacolin were used as positive controls. None of the compounds showed inhibitory effects against AChE; however, 12 compounds showed inhibitory effects against BChE. Compound **2t,** with a meta-methoxy substituent, was the most potent in the assay, with a BChE inhibition rate of 55% at 100 μM. This inhibition rate was less potent than the drug tacolin (inhibition rate: 98.2% at 100 μM). Compounds **2b** (with ortho-F) and **2l** (with meta-CF_3_) also displayed good inhibitory activity, with inhibition rates of 49.0% and 49.1%, respectively. The inhibition of ChE enzymes, namely AChE and BChE, which catalyze the hydrolysis of cholinergic neurotransmitters, may increase the levels of ACh and BCh, respectively, and remains a promising strategy for the treatment of AD [35].

### 2.5. Cellular Toxicity

Toxicity is a majorcauseofcompoundfailure at all stages of the drug development process.Mostsafety-relatedattrition occurs in the preclinical phase; therefore, safetyshouldbe predicted as early as possible during preclinical drug development. This strategy enables the design or selection of improved drug candidates that have a greater chance of becoming commercialized drugs [36]. For this reason, six derivatives with good inhibitory activity (**2d**, **2i**, **2j**, **2p**, **2t**, and **2v**) were tested for cytotoxicity. The results of the MTT cytotoxicity test are shown in Figure 4. Compared with the blank control, six compounds had a viability of more than 90%, and no obvious cytotoxicity was observed from 3.125 to 100 μM. Thus, **2d**, **2i**, **2j**, **2p**, **2t**, and **2v** have potential as drug candidates. The results of the MTT assay and AO fluorescence staining experiment were confirmed as shown in Figure 5, all six compounds showed no significant toxicity toward L929 cells at 100 μM, compared with the control group (multiples are 4×, 10×, and 20×).

### 2.6. Molecular DockingTest

Molecular modeling was performed using the Vina (Simina) docking program to further investigate the interaction mode of compound **2t** with MAO-A and MAO-B. Compound **2t**, which displaysgood MAO-A and MAO-B inhibitory activity, was selected for the molecular docking study. The interaction modes of **2t** with MAO-A and MAO-B revealed that **2t** binds to the active sites of MAO-A and MAO-B, occupying the entire enzymatic catalytic active site (CAS), mid-gorge site, and a peripheral anionic site (PAS) [37]. The docking energies of **2t** were determined to be −9.7 and −8.17 kcal/mol against MAO-A and MAO-B, respectively, as shown in Figure 6 (left: MAO-A, right: MAO-B). The cognate ligand showed conventional hydrogen bonds with Ile23 and a π-π T-shape with the phenyl ring of Tyr407in the active site of MAO-A. However, a strong hydrogen bond was also formed between the oxygen atom of the carbonyl group of the amide bondand Ser59 and Tyr60 in the active site of MAO-B.These results suggested that compound **2t** is a dual inhibitor of MAO-A and MAO-B, which is consistent with the results obtained from previous analyses.

## 3. Experimental Protocols

### 3.1. Reagents and Instruments

The mainchemical agents are bought from Aldrich ChemicalCorporation (Shanghai, China). Infrared spectra (IR in KBr) were takennote of for FT-IR1730 (Bruker, Switzerland). The nuclearmagnetic resonance spectrum was detectedon an AV-300 (Bruker, Switzerland). The shift ofchemistry was provided with ppm related totetramethylsilane. Mass spectra (MS) were detectedthrough ESI-HRMS (Brook Dalton Instruments, Germany).The absorbance value of each good sample wasdetected on the Epoch 2 microplate reader. HPLC is detected by 1220 Infinity IILC (Agilent Technologies, Santa Clara, CA, USA). An Agilent liquid chromatograph was used to detect its purity. Column temperature: 35 °C, detection wavelength: 204 nm, mobile phase: methanol: water (V:V, 82:18), chromatographic column: DIAMONSIL^TM^ C_18_ column (250 mm × 4.6 mm, 5 μm).

### 3.2. Synthesis of the Compounds

#### 3.2.1. Preparation of Substituted 1-(Isocyanatomethyl)benzene (**1a**–**1ac**)

To a stirred solution of triphosgene (0.01 mol) in dry dichloromethane was slowly added a solution of substituted benzylamine (0.02 mol) and triethylamine (0.03 mol) in dichloromethane at 0 °C. The resulting mixture was stirred at 0 °C for 30 min, then at room temperature for 1–2 h (reaction progress was monitored by TLC), and the mixture was concentrated into a residue to give intermediates **1a**–**1ac**. Compounds were purified by recrystallization with EtOH. The yield was 74−91% [38].

#### 3.2.2. Preparation of 3,4-Dihydroisoquinoline Carboxamide Derivatives **2a**–**2ac**

To a stirred solution of **1a**–**1ac** (0.02 mol) and triethylamine (0.03 mol) in dry tetrahydrofuran was slowly added (*S*)-1-phenyl-1,2,3,4-tetrahydroisoquinoline (0.02 mol), and the resulting mixture was stirred at 66 °C for 12 h (reaction progress was monitored by TLC). The mixture was then filtered, and the filtrate was concentrated to a residue, which was purified by silica gel column chromatography/recrystallization to give target compounds **2a**–**2ac** (Figure 1) [39].

### 3.3. Spectral Data of 3,4-Dihydroisoquinoline Carboxamide Derivatives ***2a**–**2ac***

(*S*)-*N*-benzyl-1-phenyl-3,4-dihydroisoquinoline-2(1*H*)-carboxamide (**2a**)

HPLC/Purity: 97.4068% (t_R_ = 7.061), Yield: 41%. mp: 122.5–123.7 °C. IR (KBr) cm^−1^: 3330, 1610, 1532, 1288. ^1^H NMR(CDCl_3,_ 300 MHz): *δ* 7.16–7.32 (m, 14H, C_6_H_5_), 6.37 (s, 1H, CH), 4.81 (t, *J* = 9.0 Hz, 1H, NH), 4.47 (d, *J* = 9.0 Hz, 2H, N-CH_2_), 3.62–3.63 (2H, m, CH_2_), 2.79–2.93 (2H, m, CH_2_); ^13^C NMR (CDCl_3_, 75 MHz): *δ* 157.44, 142.77, 139.42, 136.50, 135.06, 128.62, 128.50, 128.40, 128.25, 127.66, 127.50, 127.27, 127.18, 126.41, 57.90, 45.14, 40.22, 28.38. ESI-MS calcd for C_23_H_22_N_2_O ^+^([M + H]^+^): 343.1732; found: 343.1720.

(*S*)-*N*-(*o*-fluorobenzyl)-1-phenyl-3,4-dihydroisoquinoline-2(1*H*)-carboxamide (**2b**)

HPLC/Purity: 93.3234% (t_R_ = 7.097), Yield: 54%. IR (KBr) cm^−1^: 3302, 1613, 1534,1220. ^1^H NMR(CDCl_3,_ 300 MHz): *δ* 6.98–7.30 (m, 13H, C_6_H_5_), 6.33 (s, 1H, CH), 4.94 (t, *J* = 9.0 Hz, 1H, NH), 4.49 (d, *J* = 9.0 Hz, 2H, N-CH_2_), 3.59–3.62 (m, 2H, CH_2_), 2.80–2.88 (m, 2H, CH_2_); ^13^C NMR (CDCl_3_, 75 MHz): *δ* 162.04, 160.09, 157.43, 142.70, 136.50, 135.02, 130.22, 129.95, 128.92, 128.48, 128.37, 128.21, 127.41, 127.24, 127.17, 126.38, 124.25, 115.32, 115,23, 57.85, 40.15, 39.13, 28.34. ESI-MS calcd for C_23_H_21_FN_2_O^+^([M + H]^+^): 361.1638; found: 361.1619.

(*S*)-*N*-(*m*-fluorobenzyl)-1-phenyl-3,4-dihydroisoquinoline-2(1*H*)-carboxamide (**2c**)

HPLC/Purity: 98.7453% (t_R_ = 7.089), Yield: 55%. IR (KBr) cm^−1^: 3334, 1612, 1526, 1240. ^1^H NMR(CDCl_3,_ 300 MHz): *δ* 6.90–7.29 (m, 13H, C_6_H_5_), 6.34 (s, 1H, CH), 4.90 (t, *J* = 9.0 Hz, 1H, NH), 4.43 (d, *J* = 9.0 Hz, 2H, N-CH_2_), 3.63–3.65 (m, 2H, CH_2_), 2.81–2.90 (m, 2H, -CH_2_); ^13^C NMR (CDCl_3_, 75 MHz): *δ* 163.88,161.79, 157.42, 142.73, 142.19, 136.28, 135.01, 130.07, 128.56, 128.40, 128.20, 127.39, 127.35, 127.21, 126.45, 122.98, 122.96, 114.37, 113.97, 57.97, 44.50, 40.24, 28.42. ESI-MS calcd for C_23_H_21_FN_2_O ^+^([M + H]^+^): 361.1638; found: 361.1610.

(*S*)-*N*-(*p*-fluorobenzyl)-1-phenyl-3,4-dihydroisoquinoline-2(1*H*)-carboxamide (**2d**)

HPLC/Purity: 96.1534% (t_R_ = 7.141), Yield: 47%. IR (KBr) cm^−1^: 3324, 1616, 1500, 1213. ^1^H NMR(CDCl_3,_ 300 MHz): *δ* 6.94 -7.27 (m, 13H, C_6_H_5_), 6.34 (s, 1H, CH), 4.89 (t, *J* = 9.0 Hz, 1H, NH), 4.39 (d, *J* = 9.0 Hz, 2H, N-CH_2_), 3.60–3.63 (m, 2H, CH_2_), 2.27–2.90 (m, 2H, CH_2_); ^13^C NMR (CDCl_3_, 75 MHz): *δ* 163.00*,* 161.05, 157.44, 142.75, 136.42, 135.27, 135.01, 129.23, 129.16, 129.07, 128.50, 128.40, 128.20, 128.04, 127.97, 127.42, 127.28, 126.42, 115.43, 57.87, 44.32, 40.15, 28.37. ESI-MS calcd for C_23_H_21_FN_2_O^+^([M + H]^+^): 361.1616; found: 361.1615.

(*S*)-*N*-(*o*-chlorobenzyl)-1-phenyl-3,4-dihydroisoquinoline-2(1*H*)-carboxamide (**2e**)

HPLC/Purity: 92.0620% (t_R_ = 8.123), Yield: 67%. mp: 145.6–147.0 °C. IR (KBr) cm^−1^: 3356, 1618, 1532, 1237. ^1^H NMR(CDCl_3,_ 300 MHz): *δ* 7.15–7.35 (m, 13H, C_6_H_5_), 6.32 (s, 1H, CH), 5.07 (t, *J* = 9.0 Hz, 1H, NH), 4.52 (d, *J* = 9.0 Hz, 2H, N-CH_2_), 3.63–3.65 (m, 2H, CH_2_), 2.79–2.92 (m, 2H, CH_2_); ^13^C NMR (CDCl_3_, 75 MHz):*δ* 157.38, 142.75, 136.83, 136.49, 135.07, 133.56, 130.30, 129.41, 128.66, 128.52, 128.39, 128.21, 127.43, 127.28, 127.16, 127.04, 126.41, 57.97, 43.05, 40.22, 28.41. ESI-MS calcd for C_23_H_21_ClN_2_O^+^([M + H]^+^): 377. 1342; found: 377.1344.

(*S*)-*N*-(*m*-chlorobenzyl)-1-phenyl-3,4-dihydroisoquinoline-2(1*H*)-carboxamide (**2f**)

HPLC/Purity: 90.8954% (t_R_ = 8.344), Yield: 61%. IR (KBr) cm^−1^: 3358, 1618, 1530, 1233. ^1^H NMR(CDCl_3,_ 300 MHz): *δ* 7.10–7.29 (m, 13H, C_6_H_5_), 6.33 (s, 1H, CH), 4.84 (t, *J* = 9.0 Hz, 1H, NH), 4.42 (d, *J* = 9.0 Hz, 2H, N-CH_2_), 3.64–3.66 (m, 2H, CH_2_), 2.83–2.90 (m, 2H, CH_2_); ^13^C NMR (CDCl_3_, 75 MHz): *δ* 157.37, 142.71, 142.68, 136.41, 135.01, 134.44, 129.85, 128.59, 128.52, 128.40, 128.20, 127.55, 127.38, 127.30, 127.22, 127.21, 126.47, 126.44, 125.66, 58.01, 44.48, 40.29, 28.31. ESI-MS calcd for C_23_H_21_ClN_2_O ^+^([M + H]^+^): 377.1342; found: 377.1349.

(*S*)-*N*-(*p*-chlorobenzyl)-1-phenyl-3,4-dihydroisoquinoline-2(1*H*)-carboxamide (**2g**)

HPLC/Purity: 95.6916% (t_R_ = 8.588), Yield: 47%. IR (KBr) cm^−1^: 3346, 1612, 1532, 1230. ^1^H NMR(CDCl_3,_ 300 MHz): *δ* 7.13 -7.29 (m, 13H, C_6_H_5_), 6.33 (s, 1H, CH), 4.91(t, *J* = 12.0 Hz, 1H, NH), 4.42 (d, *J* = 12.0 Hz, 2H, N-CH_2_), 3.60–3.64 (m, 2H, CH_2_), 2.79–2.89 (m, 2H, CH_2_); ^13^C NMR (CDCl_3_, 75 MHz): *δ* 157.44, 142.75, 138.12, 136.40, 135.01, 132.94, 128.90, 128.79, 128.68, 128.55, 128.42, 128.27, 128.21, 127.43, 127.33, 127.22, 126.46, 57.94, 44.35, 40.20, 28.41. ESI-MS calcd for C_23_H_21_ClN_2_O^+^([M + H]^+^): 377.1342; found: 377.1338.

(*S*)-*N*-(*o*-bromobenzyl)-1-phenyl-3,4-dihydroisoquinoline-2(1*H*)-carboxamide (**2h**)

HPLC/Purity: 97.4021% (t_R_ = 8.545), Yield: 69%. mp: 165.9–166.6 °C. IR (KBr) cm^−1^: 3357, 1618, 1527, 1237. ^1^H NMR(CDCl_3,_ 300 MHz): *δ* 7.10–7.52 (m, 13H, C_6_H_5_), 6.32 (1H, s, CH), 5.12 (t, *J* = 9.0 Hz, 1H, NH), 4.50 (d, *J* = 9.0 Hz, 2H, N-CH_2_), 3.63–3.65 (m, 2H, CH_2_), 2.80–2.90 (m, 2H, CH_2_); ^13^C NMR (CDCl_3_, 75 MHz): *δ* 157.34, 142.75, 138.43, 136.49, 135.08, 132.67, 130.53, 130.51, 128.93, 128.52, 128.39, 128.20, 127.67, 127.43, 127.28, 127.17, 126.42, 123.72, 57.97, 45.33, 40.21, 28.41. ESI-MS calcd for C_23_H_21_BrN_2_O^+^([M + H]^+^): 421.0837; found: 421.0832.

(*S*)-*N*-(*m*-bromobenzyl)-1-phenyl-3,4-dihydroisoquinoline-2(1*H*)-carboxamide (**2i**)

HPLC/Purity: 97.5607% (t_R_ = 8.877), Yield: 42%. IR (KBr) cm^−1^: 3356, 1618, 1528, 1230. ^1^H NMR(CDCl_3,_ 300 MHz): *δ* 7.13–7.36 (m, 13H, C_6_H_5_), 6.34 (1H, s, CH), 4.97 (t, *J* = 9.0 Hz, 1H, NH), 4.42 (d, *J* = 9.0 Hz, 2H, N-CH_2_), 3.61–3.65 (m, 2H, CH_2_), 2.79–2.90 (m, 2H CH_2_); ^13^C NMR (CDCl_3_, 75 MHz): *δ* 157.40, 142.71, 136.41, 135.03, 134.38, 130.44, 130.28, 128.94, 128.70, 128.62, 128.59 (CH), 128.25, 128.22, 127.40, 127.37, 127.34, 127.06, 126.45, 122.65, 57.95, 44.40, 40.23, 28.42. ESI-MS calcd for C_23_H_21_BrN_2_O^+^([M + H]^+^): 421.0837; found: 421.0821.

(*S*)-*N*-(*p*-bromobenzyl)-1-phenyl-3,4-dihydroisoquinoline-2(1*H*)-carboxamide (**2j**)

HPLC/Purity: 95.8056% (t_R_ = 9.161), Yield: 54%. IR (KBr) cm^−1^: 3356, 1617, 1530, 1238. ^1^H NMR(CDCl_3,_ 300 MHz): *δ* 7.07–7.40 (m, 13H, C_6_H_5_), 6.33 (s, 1H, CH), 4.95 (t, *J* = 9.0 Hz, 1H, NH), 4.36 (d, *J* = 9.0 Hz, 2H, N-CH_2_), 3.63–3.65 (m, 2H, CH_2_), 2.78–2.88 (m, 2H, CH_2_); ^13^C NMR (CDCl_3_, 75 MHz): *δ* 157.44, 142.75, 138.67, 136.39, 133.81, 131.65, 131.62, 129.27, 128.94, 128.71, 128.62, 128.55, 128.43, 128.26, 127.44, 127.34, 127.22, 126.46, 121.02, 57.93, 44.38, 40.18, 28.41. ESI-MS calcd for C_23_H_21_BrN_2_O ^+^([M + H]^+^): 421.0837; found: 421.0841.

(*S*)-*N*-(*o*-(trifluoromethyl)benzyl)-1-phenyl-3,4-dihydroisoquinoline-2(1*H*)-carboxa-mide (**2k**)

HPLC/Purity: 92.1203% (t_R_ = 7.958), Yield: 75%. IR (KBr) cm^−1^:3318, 1611, 1537, 1306. ^1^H NMR(CDCl_3,_ 300 MHz): *δ* 7.17 -7.62 (m, 13H, C_6_H_5_), 6.33 (s, 1H, CH), 4.95 (t, *J* = 9.0 Hz, 1H, NH), 4.63 (d, *J* = 9.0 Hz, 2H, N-CH_2_), 3.61–3.64 (m, 2H, CH_2_), 2.79–2.92 (m, 2H, CH_2_); ^13^C NMR (CDCl_3_, 75 MHz): *δ* 157.31, 142.72, 138.02, 136.45, 132.25, 130.63, 128.52, 128.39, 128.20, 128.07, 127.83, 127.65, 127.39, 127.29, 126.44, 125.96, 125.91, 125.87, 125.68, 123.50, 57.94, 41.53, 40.19, 28.38. ESI-MS calcd for C_24_H_21_F_3_ N_2_O^+^([M + H]^+^): 411.1606; found: 411.1602.

(*S*)-*N*-(*m*-(trifluoromethyl)benzyl)-1-phenyl-3,4-dihydroisoquinoline-2(1*H*)-carbox-amide (**2l**)

HPLC/Purity: 91.9704% (t_R_ = 8.150), Yield: 52%. IR (KBr) cm^−1^: 3326, 1612, 1535, 1312. ^1^H NMR(CDCl_3,_ 300 MHz): *δ* 7.15–7.53 (m, 13H, C_6_H_5_), 6.34 (s, 1H, CH), 5.03 (t, *J* = 9.0 Hz, 1H, NH), 4.48 (d, *J* = 9.0 Hz, 2H, N-CH_2_), 3.62–3.65 (m, 2H, CH_2_), 2.81–2.89 (m, 2H, -CH_2_); ^13^C NMR (CDCl_3_, 75 MHz): *δ* 157.44, 142.65, 139.22, 136.36, 135.00, 131.23, 130.97, 129.81, 128.77, 128.42, 128.19, 127.39, 127.35, 127.25, 126.47, 125.20, 124.33, 124.06, 124.03, 123.04, 58.00, 44.79, 40.26, 28.40. ESI-MS calcd for C_24_H_21_F_3_N_2_O^+^([M + H]^+^): 411.1606; found: 411.1612.

(*S*)-*N*-(*p*-(trifluoromethyl)benzyl)-1-phenyl-3,4-dihydroisoquinoline-2(1*H*)-carboxa-mide (**2m**)

HPLC/Purity: 92.1722% (t_R_ = 8.509), Yield: 64%. IR (KBr) cm^−1^: 3349, 1617, 1532, 1319. ^1^H NMR(CDCl_3,_ 300 MHz): *δ* 7.15–7.53 (m, 13H, C_6_H_5_), 6.33 (s, 1H, CH), 5.01 (t, *J* = 9.0 Hz, 1H, NH), 4.47(d, *J* = 9.0 Hz, 2H, N-CH_2_), 3.66–3.68 (m, 2H, CH_2_), 2.80–2.93 (m, 2H, CH_2_); ^13^C NMR (CDCl_3_, 75 MHz): *δ* 156.65, 143.77, 141.31, 136.34, 134.66, 128.69, 128.44, 128.25, 128.02, 127.61, 126.97, 126.61, 126.23, 125.17, 124.89, 123.10, 58.74, 43.75, 40.25, 28.02. ESI-MS calcd for C_24_H_21_F_3_N_2_O ^+^([M + H]^+^): 411.1606; found: 411.1611.

(*S*)-*N*-(*o*-(trifluoromethoxy)benzyl)-1-phenyl-3,4-dihydroisoquinoline-2(1*H*)-carbox-amide (**2n**)

HPLC/Purity: 94.9895% (t_R_ = 8.545), Yield: 65%. IR (KBr) cm^−1^: 3301, 1614, 1535, 1248. ^1^H NMR(CDCl_3,_ 300 MHz): *δ* 7.16–7.39 (m, 13H, C_6_H_5_), 6.33 (s, 1H, CH), 4.95 (t, *J* = 9.0 Hz, 1H, NH), 4.51 (d, *J* = 9.0 Hz, 2H, N-CH_2_), 3.62–3.65 (m, 2H, CH_2_), 2.80–2.92 (m, 2H, CH_2_); ^13^C NMR (CDCl_3_, 75 MHz): *δ* 157.41, 147.35, 142.72, 136.46, 135.03, 132.01, 130.26, 128.62, 128.52, 128.39, 128.20127.39, 127.29, 127.19, 127.06, 126.43, 121.61, 120.42, 119.57, 57.95, 40.18, 39.68, 28.38. ESI-MS calcd for C_24_H_21_F_3_N_2_ O_2_^+^ ([M + H]^+^): 427.1555; found: 427.1561.

(*S*)-*N*-(*p*-(trifluoromethoxy)benzyl)-1-phenyl-3,4-dihydroisoquinoline-2(1*H*)-carbox-amide(**2o**)

HPLC/Purity: 93.1264% (t_R_ = 9.132), Yield: 57%. IR (KBr) cm^−1^: 3303, 1612, 1533, 1258. ^1^H NMR(CDCl_3,_ 300 MHz): *δ* 7.11–7.27 (m, 13H, C_6_H_5_), 6.34 (s, 1H, CH), 5.00 (t, *J* = 9.0 Hz, 1H, NH), 4.47 (d, *J* = 9.0 Hz, 2H, N-CH_2_), 3.62–3.65 (m, 2H, CH_2_), 2.81–2.93 (m, 2H, CH_2_); ^13^C NMR (CDCl_3_, 75 MHz): *δ* 157.42, 142.73, 138.38, 136.38, 134.99, 128.88, 128.56, 128.42, 128.21, 127.42, 127.35, 127.23, 126.47, 121.19, 121.11, 57.98, 44.26, 40.24, 28.42. ESI-MS calcd for C_24_H_21_F_3_N_2_O_2_^+^([M + H]^+^): 427.1555; found: 427.5544.

(*S*)-*N*-(*o*-methylbenzyl)-1-phenyl-3,4-dihydroisoquinoline-2(1*H*)-carboxamide (**2p**)

HPLC/Purity: 97.3785% (t_R_ = 8.011), Yield: 57%. mp:125.6–127.3 °C. IR (KBr) cm^−1^: 3340, 1610, 1524, 1235. ^1^H NMR(CDCl_3,_ 300 MHz): *δ* 7.03–7.26 (m, 13H, C_6_H_5_), 6.37 (s, 1H, CH), 4.78 (t, *J* = 6.0 Hz, 1H, NH), 4.42 (d, *J* = 6.0 Hz, 2H, N-CH_2_), 3.62–3.63 (m, 2H, CH_2_), 2.82–2.89 (m, 2H, CH_2_), 2.32 (3H, s, CH_3_); ^13^C NMR (CDCl_3_, 75 MHz): *δ* 157.47, 142.83, 139.33, 138.28, 136.53, 135.09, 130.44, 128.51, 128.47, 128.40, 128.38, 128.29, 128.01, 127.47, 127.23, 127.15, 126.39, 126.12, 124.68, 57.83, 45.10, 40.15, 28.39, 21.38. ESI-MS calcd for C_24_H_24_N_2_O^+^([M + H]^+^): 357.1889; found: 357.1872.

(*S*)-*N*-(*m*-methylbenzyl)-1-phenyl-3,4-dihydroisoquinoline-2(1*H*)-carboxamide (**2q**)

HPLC/Purity: 97.9738% (t_R_ = 8.001), Yield: 51%. mp:142.7–144.0 °C. IR (KBr) cm^−1^: 3346, 1610, 1525, 1240. ^1^H NMR(CDCl_3,_ 300 MHz): *δ* 7.05–7.27 (m, 13H, C_6_H_5_), 6.37 (s, 1H, CH), 4.78 (t, *J* = 9.0 Hz, 1H, NH), 4.45 (d, *J* = 9.0 Hz, 2H, N-CH_2_), 3.60–3.63 (m, 2H, CH_2_), 2.82–2.89 (m, 2H, CH_2_), 2.32 (s, 3H, CH_3_); ^13^C NMR (CDCl_3_, 75 MHz): *δ* 157.47, 142.83, 139.33, 138.28, 136.53, 135.09, 128.51, 128.47, 128.40, 128.38, 128.23, 128.01, 127.47, 127.23, 127.15, 126.39, 124.68, 57.83, 45.10, 40.15, 28.39, 21.38. ESI-MS calcd for C_24_H_24_N_2_O^+^ ([M + H]^+^): 357.1889; found: 357.1893.

(*S*)-*N*-(*p*-methylbenzyl)-1-phenyl-3,4-dihydroisoquinoline-2(1*H*)-carboxamide (**2r**)

HPLC/Purity: 94.9442% (t_R_ = 7.649), Yield: 56%. IR (KBr) cm^−1^: 3356, 1616, 1527, 1248. ^1^H NMR(CDCl_3,_ 300 MHz): *δ* 7.09–7.26 (m, 13H, C_6_H_5_), 6.37 (s, 1H, CH), 4.81 (t, *J* = 9.0 Hz, 1H, NH), 4.43 (d, *J* = 9.0 Hz, 2H, N-CH_2_), 3.57–3.61 (m, 2H, CH_2_), 2.80–2.88 (m, 2H, CH_2_), 2.30 (s, 3H, CH_3_); ^13^C NMR (CDCl_3_, 75 MHz): *δ* 157.50, 142.84, 136.93, 136.55, 136.41, 135.11, 129.29, 128.47, 128.40, 128.26, 127.70, 127.51, 127.22, 127.16, 126.39, 57.78, 44.91, 40.11, 28.37, 21.10. ESI-MS calcd for C_24_H_24_N_2_O^+^([M + H]^+^): 357.1889; found: 357.1896.

(*S*)-*N*-(*o*-methoxybenzyl)-1-phenyl-3,4-dihydroisoquinoline-2(1*H*)-carboxamid (**2s**)

HPLC/Purity: 97.2296% (t_R_ = 7.344), Yield: 68%. mp:165.5–166.6 °C. IR(KBr)cm^−1^: 3431, 1638, 1512, 1239. ^1^H NMR(CDCl_3,_ 300 MHz): *δ* 6.81–7.23 (m, 13H, C_6_H_5_), 6.31 (s,1H, CH), 5.14 (t, *J* = 6.0 Hz, 1H, NH), 4.45 (d, *J* = 6.0 Hz, 2H, N-CH_2_), 3.76 (s, 3H, OCH_3_), 3.60–3.76 (m, 2H, CH_2_), 2.72–2.91 (m, 2H, CH_2_); ^13^C NMR (CDCl_3_, 75 MHz): *δ* 157.75, 157.56, 142.92, 136.69, 135.27, 129.76, 128.97, 128.57, 128.42, 128.36, 128.19, 127.45, 127.14, 127.11, 126.35, 120.72, 110.23, 57.90, 55.26, 41.17, 40.14, 28.38. ESI-MS calcd for C_24_H_24_N_2_O_2_
^+^([M + H]^+^): 373.1838; found: 373.1822.

(*S*)-*N*-(*m*-methoxybenzyl)-1-phenyl-3,4-dihydroisoquinoline-2(1*H*)-carboxamide (**2t**)

HPLC/Purity: 96.3463% (t_R_ = 6.838), Yield: 72%. mp: 108.4–109.4 °C. IR (KBr) cm^−1^: 3342, 1611, 1528, 1257. ^1^H NMR(CDCl_3,_ 300 MHz): *δ* 6.80–7.26 (m, 13H, C_6_H_5_), 6.38 (s, 1H, CH), 4.80 (s, 1H, -NH), 4.44 (d, *J* = 6.0 Hz, 2H, N-CH_2_), 3.77 (s, 3H, OCH_3_), 3.61–3.63 (m, 2H, CH_2_), 2.79–2.90 (m, 2H, CH_2_); ^13^C NMR (CDCl_3_, 75 MHz): *δ* 159.86, 157.46, 142.78, 141.08, 136.51, 135.06, 129.63, 128.47, 128.39, 128.24, 127.47, 127.24, 127.17, 126.40, 119.86, 113.11, 112.86, 57.81, 55.22, 45.10, 40.16, 28.36. ESI-MS calcd for C_24_H_24_N_2_O_2_
^+^([M + H]^+^): 373.1838; found: 373.1843.

(*S*)-*N*-(*p*-methoxybenzyl)-1-phenyl-3,4-dihydroisoquinoline-2(1*H*)-carboxamide (**2u**)

HPLC/Purity: 96.9381% (t_R_ = 8.111), Yield: 44%. IR (KBr) cm^−1^: 3346, 1612, 1526, 1248. ^1^H NMR(CDCl_3,_ 300 MHz): *δ* 6.06–7.51 (m, 13H, C_6_H_5_), 6.47 (s, 1H, CH), 6.10 (t, *J* = 6.0 Hz, 1H, NH), 4.50 (d, *J* = 6.0 Hz, 2H, N-CH_2_), 3.77 (s, 3H, OCH_3_), 3.49–3.75 (m, 2H, CH_2_), 2.72–2.81 (m, 2H, CH_2_); ^13^C NMR (CDCl_3_, 75 MHz): *δ* 162.33, 147.70, 141.19, 139.97, 134.45, 133.26, 132.96, 132.42, 131.93, 131.79, 131.75, 130.87, 128.85, 125.32, 119.87, 119.59, 115.31, 61.93, 45.68, 44.29, 43.15, 32.96. ESI-MS calcd for C_24_H_24_N_2_O_2_
^+^([M + H]^+^): 373.1838; found: 373.1830.

(*S*)-*N*-(2,4-dimethoxybenzyl)-1-phenyl-3,4-dihydroisoquinoline-2(1*H*)-carboxamide (**2v**)

HPLC/Purity: 90.5831% (t_R_ = 7.378), Yield: 78%. mp: 153.5–155.4 °C. IR (KBr) cm^−1^: 3345, 1614, 1534, 1201. ^1^H NMR(CDCl_3,_ 300 MHz): *δ* 6.41–7.26 (m, 12H, C_6_H_5_), 6.29 (s, 1H, CH), 5.02 (t, *J* = 6.0 Hz, 1H, NH), 4.36 (d, *J* = 6.0 Hz, 2H, N-CH_2_), 3.79 (s, 3H, OCH_3_), 3.74 (s, 3H, OCH_3_), 3.55–3.60 (m, 2H, CH_2_), 2.78–2.87 (m, 2H, CH_2_); ^13^C NMR (CDCl_3_, 75 MHz): *δ* 160.33, 158.58, 157.73, 130.54, 128.38, 128.33, 128. 17, 127.44, 127.10, 127.07, 126.31, 120.03, 103.91, 98.62, 57.88, 55.42, 55.28, 40.76, 40.10, 28.36. ESI-MS calcd for C_25_H_26_N_2_O_3_^+^([M + H]^+^): 403.1943; found: 403.1976.

(*S*)-*N*-(*3,4*-dimethoxybenzyl)-1-phenyl-3,4-dihydroisoquinoline-2(1*H*)-carboxamide (**2w**)

HPLC/Purity: 97.5223% (t_R_ = 6.851), Yield: 53%. IR (KBr) cm^−1^: 3335, 1612, 1514, 1216. ^1^H NMR(CDCl_3,_ 300 MHz): *δ* 6.95–7.26 (m, 12H, C_6_H_5_), 6.48 (s, 1H, CH), 4.99 (t, *J* = 6.0 Hz, 1H, NH), 4.19 (d, *J* = 6.0 Hz, 2H, N-CH_2_), 3.78 (s, 3H, OCH_3_), 3.74 (s, 3H, OCH_3_), 3.19–3.16 (m, 2H, CH_2_), 2.61–2.69 (m, 2H, CH_2_); ^13^C NMR (CDCl_3_, 75 MHz): *δ* 158.09, 154.92, 148.97, 148.29, 141.27, 133.60, 130.75, 129.72, 128.01, 127.28, 125.91, 120.32, 119.28, 111.91, 111.23, 58.06, 55.95, 44.79, 40.25, 28.02. ESI-MS calcd for C_25_H_26_N_2_O_3_^+^ ([M + H]^+^): 403.1943; found: 403.1935.

(*S*)-*N*-(*2,4*-dichlorobenzyl)-1-phenyl-3,4-dihydroisoquinoline-2(1*H*)-carboxamide(**2x**)

HPLC/Purity: 95.9287% (t_R_ = 11.059), Yield: 64%.mp: 130.8–132.7 °C. IR (KBr) cm^−1^: 3280, 1612, 1517, 1210. ^1^H NMR(CDCl_3,_ 300 MHz): *δ* 7.15–7.33 (m, 12H, C_6_H_5_), 6.28 (s, 1H, CH), 5.10 (s, 1H, -NH), 4.48 (d, *J* = 6.0 Hz, 2H, N-CH_2_), 3.63–3.65 (m, 2H, CH_2_), 2.79–2.92 (m, 2H, CH_2_); ^13^C NMR (CDCl_3_, 75 MHz): *δ* 157.32, 142.66, 136.35, 135.52, 134.98, 134.05, 133.63, 131.03, 129.17, 128.58, 128.42, 128.18, 127.39, 127.26, 127.22, 126.47, 58.10, 42.46, 40.29, 28.42. ESI-MS calcd for C_23_H_20_Cl_2_N_2_O^+^([M + H]^+^): 411.0953; found: 411.0958.

(*S*)-*N*-(*2,6*-dichlorobenzyl)-1-phenyl-3,4-dihydroisoquinoline-2(1*H*)-carboxamide(**2y**)

HPLC/Purity: 98.4243% (t_R_ = 9.300), Yield: 68%. mp: 169.0–169.4 °C. IR (KBr) cm^−1^: 3289, 1617, 1527, 1220. ^1^H NMR(CDCl_3,_ 300 MHz): *δ* 7.13–7.30 (m, 12H, C_6_H_5_), 6.29 (s, 1H, CH), 4.96 (t, *J* = 6.0 Hz, 1H, NH), 4.74 (d, *J* = 6.0 Hz, 2H, N-CH_2_), 3.60–3.64 (m, 2H, CH_2_), 2.76–2.91 (m, 2H, CH_2_); ^13^C NMR (CDCl_3_, 75 MHz):*δ* 157.16, 142.75, 136.49, 136.14, 135.10, 134.67, 129.28, 128.49, 128.39, 128.21, 127.45, 127.26, 127.11, 126.36, 57.98, 40.72, 40.18, 28.41. ESI-MS calcd for C_23_H_20_Cl_2_N_2_O ^+^([M + H]^+^): 411.0953; found: 411.0962.

(*S*)-*N*-(*3,4*-dichlorobenzyl)-1-phenyl-3,4-dihydroisoquinoline-2(1*H*)-carboxamide (**2z**)

HPLC/Purity: 97.6748% (t_R_ = 10.458), Yield: 67%. mp: 120.1–120.9 °C. IR (KBr) cm^−1^: 3282, 1615, 1523, 1224 ^1^H NMR(CDCl_3,_ 300 MHz): *δ* 7.04–7.33 (m, 12H, C_6_H_5_), 6.31 (s, 1H, CH), 5.03 (s, 1H, NH), 4.41 (d, *J* = 9.0 Hz, 2H, N-CH_2_), 3.64–3.67 (m, 2H, CH_2_), 2.79–2.92 (m, 2H, CH_2_); ^13^C NMR (CDCl_3_, 75 MHz): *δ* 157.35, 142.64, 140.06, 136.30, 134.95, 132.52, 131.06,130.46, 129.28, 128.64, 128.44, 128.18, 127.44, 127.36, 127.27, 126.81, 126.50, 58.04, 43.87, 40.28, 28.43. ESI-MS calcd for C_23_H_20_Cl_2_N_2_O ^+^ ([M + H]^+^): 411.0953; found: 411.0942.

(*S*)-*N*-(*2,4*-difluorobenzyl)-1-phenyl-3,4-dihydroisoquinoline-2(1*H*)-carboxamide(**2aa**)

HPLC/Purity: 95.3932% (t_R_ = 7.433), Yield: 47%. IR (KBr) cm^−1^: 3303, 1612, 1525, 1223. ^1^H NMR(CDCl_3,_ 300 MHz): *δ* 7.15–7.27 (m, 12H, C_6_H_5_), 6.31 (s, 1H, CH), 4.97 (t, *J* = 9.0 Hz, 1H, NH), 4.43(d, *J* = 9.0 Hz, 2H, N-CH_2_), 3.61–3.64(m, 2H, CH_2_), 2.72–2.91 (m, 2H, CH_2_); ^13^C NMR (CDCl_3_, 75 MHz): *δ* 163.26, 163.16, 161.98, 161.89, 161.28, 161.19, 159.92, 157.40, 155.04, 142.68, 136.41, 135.01, 128.63, 128.51, 128.40, 128.21, 128.07, 127.40, 127.30, 127.21, 126.58, 111.29, 103.19, 57.90, 39.87, 38.45, 28.69. ESI-MS calcd for C_23_H_20_F_2_ N_2_O ^+^ ([M + H]^+^): 379.1544; found: 379.1552.

(*S*)-*N*-(*2,6*-difluorobenzyl)-1-phenyl-3,4-dihydroisoquinoline-2(1*H*)-carboxamide(**2ab**)

HPLC/Purity: 94.9327% (t_R_ = 6.820), Yield:71%. mp: 125.3–128.4 °C. IR (KBr) cm^−1^: 3301, 1615, 1524, 1228. ^1^H NMR(CDCl_3,_ 300 MHz): *δ* 7.84–7.25 (m, 12H, C_6_H_5_), 6.32 (s, 1H, CH), 4.92 (s, 1H, -NH), 4.52 (d, *J* = 9.0 Hz, 2H, N-CH_2_), 3.58–3.60 (m, 2H CH_2_), 2.75–2.90 (m, 2H, CH_2_); ^13^C NMR (CDCl_3_, 75 MHz): *δ* 162.61, 162.54, 160.63, 160.57, 157.14, 142.68, 136.49, 135.08, 129.13, 128.44, 128.37, 128.23, 127.45, 127.21, 127.13, 126.36, 115.08, 114.93, 111.33, 57.79, 40.08, 33.13, 28.32. ESI-MS calcd for C_23_H_20_F_2_N_2_O^+^ ([M + H]^+^): 379.1544; found: 379.1548.

(*S*)-*N*-(*p*-cyanobenzyl)-1-phenyl-3,4-dihydroisoquinoline-2(1*H*)-carboxamide(**2ac**)

HPLC/Purity: 98.0928% (t_R_ = 5.802), Yield:49%. IR (KBr) cm^−1^: 3329, 1620, 1526, 1232. ^1^H NMR(CDCl_3,_ 300 MHz): *δ* 7.16–7.56 (m, 13H, C_6_H_5_), 6.31 (s, 1H, CH), 5.04 (t, *J* = 3.0 Hz, 1H, NH), 4.44 (d, *J* = 3.0 Hz, 2H, N-CH_2_), 3.65–3.68 (m, 2H, CH_2_), 2.84–2.98 (m, 2H, CH_2_); ^13^C NMR (CDCl_3_, 75 MHz): *δ* 157.37, 145.34, 142.66, 136.24, 134.91, 132.40, 132.34, 128.79, 128.69, 128.46, 128.28, 128.07, 127.89, 127.45, 127.39, 127.30, 126.54, 118.86, 110.89, 58.08, 44.50, 40.30, 28.44. ESI-MS calcd for C_24_H_21_N_3_O^+^([M + H]^+^): 368.1685; found: 368.1676.

## 4. Biological Activity

### 4.1. MAO Activity Assay

The enzyme activity analysis was performed according to the procedure reported by Matsumoto et al. with slight modifications. All materials used in the enzymatic assays were purchased from Aladdin Reagent Chemicals. A fixed concentration of tyramine substrate and different concentrations of target compounds or inhibitor were used to confirm the percentage inhibition rate or IC_50_ values. Tyramine concentrations for MAO-A and -B were 120 µM. The concentrations of the synthesized compounds altered from 0.001 µM to 100 µM for the MAO-A and -B enzyme activity inhibition. The compounds **2a**–**2ac** were dissolved in 2% DMSO, diluted in buffer solution before the test assay, and pre-cultured with the enzyme at 37 °C for 30 min. The final concentration of DMSO in the enzyme-measured reaction mixture did not exceed 1%. The enzymatic reactions were caused by the addition of MAO-A (40 µg/mL) or MAO-B (20 µg/mL), and cultured at 37 °C for 30 min. The enzyme reactions were stopped through the addition of 2N NaOH for 75 µL. Fluorescence was recorded from the top by using the Infinite 200 PRO multimode microplate reader at the excitation or emission wavelengths of 310 and 380 nm, IC_50_ values using the software Prism 8 program. The inhibitory effect of enzyme activity was calculated as a percentage of product formation compared to the corresponding control with no inhibitor. A control was established to determine the interference of the tested compounds to the fluorescence assay, and the enzyme or substrate was added after the reaction stopped.

### 4.2. In Vitro Cholinesterase Inhibition Assay

Inhibition of AChE and BChE was measured through the spectrophotometric method developed by Ellman’s method with slight modifications [34]. The analysis protocol contained 96-well plates (100 µL per well). Each well involved 20 µL of assay buffer solution, 20 µL of the test sample, and 40 µL of 0.2 U/mL of AChE or BChE. The mixture was incubated for 10 min at 25 °C followed by the addition of substrate (20 µL). For AChE inhibition assay, acetylthiocholine iodide (ATCI, 0.001 mol/L), while for BChE assay, butyrylthiocholine chloride (BTCCl, 0.001 mol/L) was added to it. The mixture was incubated at 37 °C for 15 min. Ellman reagent (5,5′-dithio-bis(2-nitrobenzoic acid), DTNB, 0.001 mol/L) was added. The change in color of the mixture displayed an indication of inhibition. The absorbance was measured at 412 nm using the microplate reader Infinite 200 PRO, Inc. Switzerland. The blank analysis was performed. All the analyses were performed in triplicate. Inhibition rate of experimental sample or positive control group (%) = [(control group-blank background)-(experimental sample or positive control group -sample background)]/(control-blank background) × 100%.

### 4.3. Cytotoxicity Test

L929 cells in the logarithmic growth phase were selected, and the cell density was adjusted to 2 × 10^4^ per well by a cell counter. The 96-well plate was inoculated in a 96-well plate, and the 96-well plate inoculated with cells was cultured in a 37 °C 5% carbon dioxide cell incubator for 24 h. Then, the medium containing 100 μM of the compound to be tested was added (the blank group was added to the medium without drug), shaken gently, and placed in a 5% carbon dioxide cell incubator at 37 °C for 24 h. Each concentration was repeated for 6 h.

#### 4.3.1. MTT Assay to Detect Cell Viability

The assay detects live cells, but not dead cells, and the resulting signal depends on how activated the cells are. Since yellow MTT can be reduced to blue-violet form azan by mitochondrial dehydrogenase in living cells, dimethyl sulfoxide can dissolve formazan (Formazan) in cells, and its light absorption value was measured with a microplate reader at a wavelength of 490 nm. Therefore, MTT has been used to establish quantitative colorimetric assays for mammalian cell survival and proliferation. Therefore, this method can be used to measure cytotoxicity, proliferation, or activation. After 24 h, 20 μL of 5 mg/kg MTT solution (prepared in PBS with pH 7.6) was added in the dark, and cultured in a cell incubator for another 4 h. After 4 h, the 96 plate was taken out to discard the original medium, and 160 μL of DMSO was added, placed on a shaker, and mixed for about 10 min. Then, a microplate reader was used to measure its absorbance at 490 nm, and the results were recorded and analyzed.

#### 4.3.2. Analysis of Cell Viability by AO Fluorescent Staining

Using the AO Fluorescent Staining Kit, AO is a tricyclic heteroaromatic dye that can penetrate cells with intact cell membranes and embed nuclear DNA, making them emit bright green fluorescence. L929 cells were selected from the logarithmic growth phase, and the cell density was adjusted to 8 × 10^5^ per well with a cell counter to inoculate in a 6-well plate, and the 6-well plate was placed in a 37 °C 5% carbon dioxide cell incubator for 12 h. It grew completely adhering to the wall. After 12 h, 1 mL of medium containing 100 μM of the compound to be tested was added (for the blank group, drug-free medium was added), shaken gently, and placed in a 5% carbon dioxide cell incubator at 37 °C for 12 h. After 12h, the medium was discarded, PBS was added to wash once, 1 mL of PBS and 80 μL of AO solution (prepared according to the kit instructions) were slowly added in the dark, and mixed for 5 min to make the staining uniform and sufficient. The staining was aspirated after 5 min, washed 1–2 times with PBS, and observed under a fluorescence microscope, the results were recorded and analyzed.

### 4.4. Molecular Docking Study

Molecular simulations and docking experiments were performed using the Vina (Simina) docking program. The protein crystal structure of human MAO-A was obtained from the Protein Crystal Database (PDB ID: 2Z5X) with a resolution of 5 Å (10–10 m). The initial structure of the MAO-A protein crystal was processed with default parameters, with the pocket being the center of the FAD ligand. Molecular simulation of the optimized protein crystal structure with 3,4-dihydroisoquinolinecarboxamide compounds **2t** and **2d** provides ideas for the design of better MAO-A inhibitors in the future.

## 5. Conclusions

In recent years, MAO and ChE inhibitors have received increased attention because of their beneficial effects on mental health. For this reason, we designed and prepared a family of (*S*)-1-phenyl-3,4-dihydroisoquinoline-2(1*H*)-carboxamide analogs. Six analogs (**2d**, **2i**, **2j**, **2p**, **2t**, and **2v**) showed good inhibitory activities against MAO with inhibition rates from 52.0 to 71.8%. Analogs **2i**, **2p**, **2t**, and **2v**, incorporating *m*-Br, *o*-CH_3_, *m*-OCH_3_, and 2,4-(OCH_3_)_2_ groups on the benzyl ring, respectively, showed potent inhibitory effects against MAO-A and MAO-B enzymes. Compounds **2d** and **2j** with *p*-F and *p*-Br groups on the benzyl ring, respectively, selectively inhibited MAO-A, with IC_50_ values of 1.38 and 2.48 µM, respectively. None of the synthesized compounds showed inhibitory activity against AChE, and 12 compounds displayed inhibitory activities against BChE. In addition, the active compounds did not show any cytotoxicity at the dose required for MAO and ChE inhibition. Our results indicated that **2t** is a potent inhibitor of MAO-A, MAO-B, and BChE enzymes, and could be a promising candidate for preclinical development for the treatment of depression, AD, and PD.

## Data Availability

The data presented in this study and associated additional data are available upon request.

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
