# Peer review of "(S)-N-Benzyl-1-phenyl-3,4-dihydroisoqunoline-2(1H)-carboxamide Derivatives, Multi-Target Inhibitors of Monoamine Oxidase and Cholinesterase: Design, Synthesis, and Biological Activity"

_molecules, 2023, doi:10.3390/molecules28041654_

Round 1

Reviewer 1 Report

Many thanks for giving me the opportunity to review this article.

In this work, the authors discuss the synthesis of a novel series of (S)-1-phenyl-3,4-dihydroisoquinoline-2(1H)-carboxamide derivatives and their evaluation as monoamine oxidase (MAO)-A and-B, acetylcholine esterase (AChE), and butyrylcholine esterase (BuChE) inhibitors. Compounds 2i, 2p, 2t, and 2v exhibited strong inhibitory activity against both MAO-A and MAO-B, while compounds 2d and 2j shown selective inhibitory activity against MAO-A, with IC50 values of 1.38 M and 2.48 M, respectively. None of the studied compounds demonstrated AChE inhibition. The target compounds exhibited no cytotoxicity against L929 cells.

The authors should consider the following points

·        The English language must be improved and refined throughout the manuscript.

·        The title of this article is (S)-N-Benzyl-1-Phenyl-3,4-Dihydroisoqunoline-2(1H)- Carboxamide Derivatives and as shown from the Scheme the authors used  benzyl isocyanate  not phenyl isocyanate and they give IUPAC names for all compounds with phenyl moiety not benzyl, So they should revised all names, data, respective ESI-MS,

·        only compound 2a is IUPAC corrected.

·         The authors should revise Experimental data very carefully and revised the manuscript as well.

Moreover,

·        In the discussion and results section, the authors described that the methylene group of the benzyl moiety (Ph-CH2) appeared as a triplet at 4.35-4.77 ppm. However, it appeared in the spectra as a doublet (d) and once as a doublet doublet (dd), needing the authors to revise the experimental data precisely and unite them with the discussed results.

·        The authors should write J-coupling constant in case of d, dd, and t.

·        The authors should revise the molecular formulas and respective ESI-MS values for all compounds. Example of compound

-(S)-N-benzyl-1-phenyl-3,4-dihydroisoquinoline-2(1H)-carboxamide (2a) has C23H23N2O, but the corrected formula is C23H22N2O, and the ESI-MS should be adjusted to reflect the correct molecular formula.

·        The synthesized compounds configurated as (S) and authors identified them by wedged bond for the chiral center, however it should be appeared as hashed wedged bond. Because  (R) recognized by wedged bond and (S) by hashed wedged bond

·        The label in supplementary file should be reported in English language  not Chinese.

The manuscript is not suitable for publication in Molecules, the authors encourage  for resubmission after carefully and precisely revision.

Kind regards

Author Response

Review 1:

  1. The English language must be improved and refined throughout the manuscript.

Answer: we revised our manuscript by Liwen Bianji (Edanz) (www.liwenbianji.cn) for editing the English text of a draft of this manuscript.

  1. The title of this article is (S)-N-Benzyl-1-Phenyl-3,4-Dihydroisoqunoline-2(1H)-Carboxamide Derivatives and as shown from the Scheme the authors used  benzyl isocyanate  not phenyl isocyanate and they give IUPAC names for all compounds with phenyl moiety not benzyl, So they should revised all names, data, respective ESI-MS, only compound 2a is IUPAC corrected.

Answer: Thank you very much, we revised all compounds names, data, and ESI-MS.

  1. The authors should revise Experimental data very carefully and revised the manuscript as well.

Answer: we revised all experimental data in the article.

  1. In the discussion and results section, the authors described that the methylene group of the benzyl moiety (Ph-CH2) appeared as a triplet at 4.35-4.77 ppm. However, it appeared in the spectra as a doublet (d) and once as a doublet doublet (dd), needing the authors to revise the experimental data precisely and unite them with the discussed results.

Answer: I'm sorry it was a mistake, we revised this parts in the article.

  1. The authors should write J-coupling constant in case of d, dd, and t.

Answer: we added to this part in the article.

  1. The authors should revise the molecular formulas and respective ESI-MS values for all compounds. Example of compound: -(S)-N-benzyl-1-phenyl-3,4-dihydroisoquinoline- 2(1H)- carboxamide (2a) has C23H23N2O, but the corrected formula is C23H22N2O, and the ESI-MS should be adjusted to reflect the correct molecular formula.

Answer: Thank you very much, we revised molecular formula of all compounds in the article.

  1. The synthesized compounds configurated as (S) and authors identified them by wedged bond for the chiral center, however it should be appeared as hashed wedged bond. Because (R) recognized by wedged bond and (S) by hashed wedged bond.

Answer: Thank you very much, we revised in the article.

  1. The label in supplementary file should be reported in English language not Chinese.

Answer: we revised this part in supplementary.

Reviewer 2 Report

The manuscript “(S)-N-Benzyl-1-Phenyl-3,4-dihydroisoquinoline-(2(1H)-carboxamide Derivatives, Multi-Target Inhibitors of Monoamine Oxidase and Cholinesterase: Design, Synthesis and Biological Activity” prepared by Q.-H. Jin and co-workers deals with the synthesis and study on biological activity of quinoline-based carboxamides bearing substituted benzyl moiety. In this study 29 novel compounds were prepared using a simple two-step procedure. Subsequently, in vitro inhibitory activity of prepared compounds against momoamine oxidase A and B, acetylcholine and butyrylcholin esterase, respectively, was assayed. Some of prepared compounds showed inhibitory activity against tested targets in micromolar range. Moreover, the most active compounds were tested on their cytotoxicity and in the case of compound 2t molecular docking study was also performed. The topic of this manuscript could be important and of interest to researchers dealing with the synthesis and/or medicinal chemistry of quinoline-based derivatives or with the design of potential therapeutics of neurodegenerative diseases. The manuscript is written intelligibly, but some of its parts are too brief, e.g. chapter 2.1 Design of analogs, or should be improved (please, see my comments in the attached file).

Despite all of comments mentioned in my review, I think that manuscript could be, after its major revision, accepted for the publication in Molecules.

Author Response

Review 2:

The manuscript is written intelligibly, but some of its parts are too brief, e.g. chapter 2.1 Design of analogs, or should be improved (please, see my comments below).

  • The numbering of compounds 1–4 in Figure 1 is confusing, because there are two series of compounds 1 and 2, respectively, in the manuscript. I suggest to insert the original names of the compounds to the Figure 1, instead the numbers. Corresponding parts of Introduction should be rewrite in the same manner.

Answer: we checked and revised in the article.

  • On page 3, compound 5 is mentioned (as the lead compound), but I did not find this compound in the manuscript. Please, explain this discrepancy.

Answer: we added the structure of lead compound in the Figure 2.

  • Chapter 2.1 “Design of analogs” should be improved, because the motivation of the authors for the design of the final compounds is not entirely clear. Were the authors inspired during the design of the prepared compounds in the literature? Is there any structurally similar compound inhibiting the same biological targets? If so, please, list and cite it. It is not clear, why the benzylamine motif was chosen and introduced into the structure of prepared compounds? (I recommend recently published comprehensive review published by Manzoor and Hoda: https://doi.org/10.1016/j.ejmech.2020.112787). Moreover, in my opinion: i) the word “carboxyamine” should be replaced with“carboxamide” and ii) in this functionality there are two places able to form hydrogen bonding–one donor and one acceptor (not only one as it is described in the manuscript).

Answer: we revised design of analogs in the article.

4) Scheme 1 (page 4) should be improved: i) molar ratios (XY equiv.) of the reactants, used solvents, temperatures and time of the reactions should be added; ii) yields of prepared compounds should be added; iii) into the legend below the scheme, compounds 1 should be added (e.g. 1a,2a=H).

Answer: we checked and appropriately revised in Scheme.

5) I disagree with the authors statement that the methylene protons of the benzyl moiety (–NH– CH2–Ph) can be appeared in the 1H NMR spectra as a triplet (page 4) – moreover, in the experimental part these signals are described as “s”, “d”, “dd” or “m” and the specified range (4.35–4.77 ppm) does not match in all cases.

Answer: I'm sorry it was a mistake, we revised this parts in the article.

6) In the discussion of the NMR spectra there is incorrectly mentioned compound 2w (it should be compound 2v).

Answer: I'm sorry it was a mistake, we revised in the article.

7) I recommend adding an illustrative figure of NMR spectra (e.g. 1H NMR of compound 2v) into the manuscript.

Answer: The NMR spectra of compounds have been provided in the supplementary materials, and the structure of compounds is relatively simple. We think that the NMR spectra of compound v need not be provide in the article.

8) There is no explanation why the rasagiline was used as a model (comparative) compound in the case of inhibitory activity against MAO (Table 1).

Answer: we revised and added to it in the article.

9) Did authors try to perform docking test for compounds 2d and 2j, respectively, selectively inhibiting MAO-A? It could be interesting to compare the results of their possible orientation and docking energy with the experiment- ally obtained values of the inhibitory activity.

Answer: thanks, your suggestion is very good. Because of the problem of the volume in the paper, we perform docking the best compound 2t that inhibits the activity MAO-A and MAO-B in this article

10) I have several reservations for “Experimental part”: i) detailed description of IR, NMR and MS experimental conditions and parameters are missing; ii) are reaction procedures described for the synthesis of compounds 1 and 2 (chapter 3.1.1. and 3.1.2.) new? If not, please, state this fact and add the citation for corresponding literature (e.g. Compounds 1a–1ac were prepared following slightly modified previously published procedure (ref)).; iii) did you isolate compounds 1a–1ac? If so, add the information about the yields.; iv) Are these compounds described in the literature? If yes, please, state whether the spectral data of compounds 1a–1ac are matched the literature; v) put the yields on a whole number (e.g. 41%); vi) in the NMR listing add the coupling constant (e.g. J = 6.8 Hz) for the signals assigned as “d”, “dd” or “t” in the 1H-NMR spectra; vii) hydrogens of methylene group of the phenyl ring should be in the 1H-NMR listing distinguished from CH2 signals of isoquinoline ring (e.g. as “-CH2Ph”).

Answer: we checked, added, and appropriately revised these parts in the article.

11) The Supporting Information file needs to be improved. Please give one spectrum per page, and have the pages in landscape orientation to make it more visible. Moreover, please, provide zoomed images for all regions where the peaks are not clearly visible. Finally, merge the Supplementary Information into one file containing both 1H- and 13CNMR spectra and HPLC chromatograms.

Answer: we checked and appropriately revised these parts in the supporting information file.

12) Typographical errors: i) unify the use of the abbreviation “BuChe” vs “BChe”; ii) replace term “scafold” for “scaffold” (Figure 2); iii) replace “(% of contron)” for“control” and “Concentration (mM)” for “ M” (Figure 4); iii) formulation “selectivity inhibited” should be change for “selectively inhibited” (page 5); formulation “no significantcy toxicity” should be change for “no significant toxicity” (page 7); sentence“These results suggest that the compound 2t are dual inhibitors…” should be written in singular, i.e. “compound 2t is a dual inhibitor…” (page 8); iv) chapter 2.2.3. should be 2.3.2.

Answer: we checked and changed it in the article.

Round 2

Reviewer 1 Report

Dear Editor of Molecules

Many thanks again, the authors have replied to all required points.

kind regards

Author Response

Dear editor,

Thank you for your review our article.

I would like to thank you for your kind consideration to publish this paper. 

Thank you for your consideration.
Sincerely yours,

Li-Ping Guan

Reviewer 2 Report

The authors revised the manuscript as recommended so that its overall quality was improved. Nevertheless, the manuscript contains some parts that should be further improved before the manuscript will be published. Please, see my comments bellow.

I think that the manuscript can be published in Molecules after its minor revision.

1)    Sentence “The C-NH signal emerged comparatively up-field as a triplet (J = 5.5 Hz) instead of a singlet in compounds.” is confusing for me. Why the C-NH signal should be appeared as singlet? Can authors explain this statement and possibly modify the wording of the sentence? 

2)   Moreover, the coupling constant reported in the chapter 2.3. (J = 5.5 Hz) is inconsistent with the “J values” of C–NH signal reported in the Experimental section, where it is typically 9.0 Hz. Can authors explain this discrepancy?

3)   It is still not clear why the rasagiline was used as a model (comparative) compound in the case of inhibitory activity against MAO. The structure of this compound is not similar to that of tested compounds. So, why authors used just this compound as a model structure?

4) I still have several reservations for “Experimental part”: i) are compounds 1a1ac described in the literature? If so, please, state whether the spectral data of compounds 1a1ac are matched the literature (with the appropriate references); ii) report the yields to a whole number (e.g. 41% for compound 2a) – the current format used by the authors is not accepted in the field of organic chemistry.

5)  In the Supporting Information file, the authors report the structure of the prepared compounds in the form of “R” instead of “S” enantiomers. It should be changed.

6)  Typographical errors: i) replace terms “liker” for “linker” (Figure 2); ii) replace “70 % – 85 % yield” for “70–85% yield” (line 147) and similarly in the line 344 (“the yields were 74–91%”).

Author Response

Dear editor,

Thank you for your review our article. We have revised our article according to the reviewer comment.

I would like to thank you for your kind consideration to publish this paper. 

Thank you for your consideration.
Sincerely yours,

Li-Ping Guan

  • Sentence “The C-NH signal emerged comparatively up-field as a triplet (= 5.5 Hz) instead of a singlet in compounds.” is confusing for me. Why the C-NH signal should be appeared as singlet? Can authors explain this statement and possibly modify the wording of the sentence?

Answer: we checked and revised it in the article.

  • Moreover, the coupling constant reported in the chapter 2.3. (J= 5.5 Hz) is inconsistent with the “J values” of C–NH signal reported in the Experimental section, where it is typically 9.0 Hz. Can authors explain this discrepancy?

Answer: I am sorry, that was a mistake, “J values” of C–NH signal is 9.0 Hz, we checked and revised it in the article.

  • It is still not clear why the rasagiline was used as a model (comparative) compound in the case of inhibitory activity against MAO. The structure of this compound is not similar to that of tested compounds. So, why authors used just this compound as a model structure?

Answer: Because rasagiline is MAOI drug, the synthesized target compounds were evaluated inhibitory activity against MAO, the aim of our experiment was to compare the inhibitory effects of the synthesized compound with that of rasagiline, so rasagiline was used as a positive controls.

4) I still have several reservations for “Experimental part”: i) are compounds 1a1ac described in the literature? If so, please, state whether the spectral data of compounds 1a1ac are matched the literature (with the appropriate references);

Answer: There is a similar literature on the synthesis of compounds 1a–1ac, we have introduced in the article. The same as compound 1a-1c has not been reported. We think that it do not to provide spectral data for compounds 1a–1ac. The target compounds 2a-2ac provide spectral data.

  1. ii) report the yields to a whole number (e.g. 41% for compound 2a) – the current format used by the authors is not accepted in the field of organic chemistry.

Answer: we checked and revised it in the article.

  • In the Supporting Information file, the authors report the structure of the prepared compounds in the form of “R” instead of “S” enantiomers. It should be changed.

Answer: we revised it In the Supporting Information file.

6)  Typographical errors: i) replace terms “liker” for “linker” (Figure 2); ii) replace “70 % – 85 % yield” for “70–85% yield” (line 147) and similarly in the line 344 (“the yields were 74–91%”).

Answer: we checked and revised it in the article.
